# Graph Differentiable Architecture Search with Structure Learning

**Yijian Qin, Xin Wang,**[*] **Zeyang Zhang, Wenwu Zhu**[*]
Tsinghua University
qinyj19@mails.tsinghua.edu.cn, xin_wang@tsinghua.edu.cn
zy-zhang20@mails.tsinghua.edu.cn, wwzhu@tsinghua.edu.cn

## Abstract

Discovering ideal Graph Neural Networks (GNNs) architectures for different tasks is labor intensive and time consuming. To save human efforts, Neural Architecture Search (NAS) recently has been used to automatically discover adequate GNN architectures for certain tasks in order to achieve competitive or even better performance compared with manually designed architectures. However, existing works utilizing NAS to search GNN structures fail to answer the question *How NAS is able to select the desired GNN architectures*. In this paper, we investigate this question to solve the problem, for the first time. We conduct theoretical analysis and measurement study with experiments to discover that gradient based NAS methods tend to select proper architectures based on the usefulness of different types of information with respect to the target task. Our explorations further show that gradient based NAS also suffers from noises hidden in the graph, resulting in searching suboptimal GNN architectures. Based on our findings, we propose a Graph differentiable Architecture Search model with Structure Optimization (GASSO), which allows differentiable search of the architecture with gradient descent and is able to discover graph neural architectures with better performance through employing graph structure learning as a denoising process in the search procedure. Extensive experiments on real-world graph datasets demonstrate that our proposed GASSO model is able to achieve the state-of-the-art performance compared with existing baselines.

## 1 Introduction

In real-world applications including social networks, e-commerce, traffics, and biochemistry, a variety of relational data can be represented as graphs. This motivates the advent of graph neural networks (GNNs) based models such as GCN [1], GAT [2] and GIN [3], which are designed to learn and extract knowledge from the relational graph-structured data. To utilize information in graph structure and node features, GNNs follow a recursive message passing scheme where nodes aggregate information from their neighbors in each layer, making great success in various graph related tasks. Given that discovering ideal GNN architectures for different tasks is labor intensive and time consuming, graph architecture search [4, 5] employs the ideas of neural architecture search (NAS) [6–8] to facilitate the automatic design of optimal GNN architectures so that a large number of human efforts can be saved. These automatically generated GNNs can achieve competitive or even better performance compared with manually designed GNNs on graph related tasks. [2]

Nevertheless, existing works on graph architecture search mainly focus on designing search space and search strategy, ignoring the fundamental mechanism in adopting NAS for automatic GNN

---

[*]Corresponding Authors.
[2]Our code will be released at `https://github.com/THUMNLab/AutoGL`

35th Conference on Neural Information Processing Systems (NeurIPS 2021), virtual.

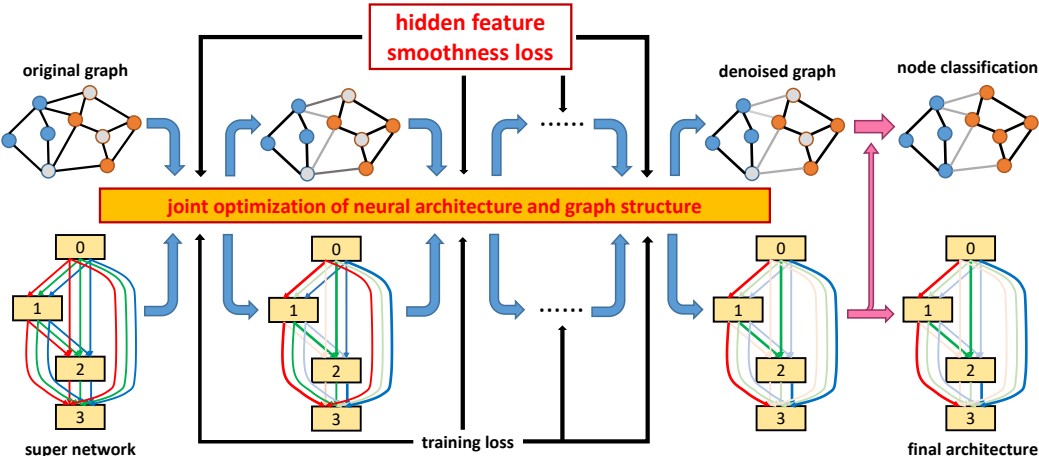

Figure 1: Overview framework of the proposed GASSO model, where the original graph and super network architecture are jointly optimized, hidden feature smoothness loss and training loss are proposed to optimize the neural architecture and graph structure. Then the learned architecture is adopted on the denoised graph, giving predictions of target nodes.

structure search and failing to answer the following questions. (i) *How does graph architecture search select its desired architectures?* and (ii) *How optimal are the architectures selected by graph neural architecture search?* Answers to these questions can help us to understand the NAS and GNN mechanism in gathering messages, leading us to design better GNN structures for certain tasks. However, the failure of existing works in answering the questions significantly limits their capabilities of designing powerful GNN architectures.

In this paper, we answer the above question through exploring how graph neural architecture search is able to select the desired GNN architectures. Given that neural architecture search approaches in literature can be mainly categorized into several groups, i.e., gradient based (DARTS [8]), reinforcement learning based, evolution algorithm based, and Bayesian optimization based methods, we only focus on gradient based architecture search approach in this work and leave investigations into the other groups as future works. We theoretically analyze DARTS behavior and find that DARTS prefers operations who can help to correct the predictions on hard data. Further analysis on graph data shows that different operations fit graphs with different amount of information in the node features and graph structures. Measurement study via designing synthetic experiments corroborates our theory. We find that DARTS on GNN is able to evaluate the usefulness of information hidden in node features and graph structure with respect to the target task, then automatically select appropriate operations based on the evaluated usefulness for GNN architecture. On the other hand, we also discover that the performance of gradient based graph architecture search (e.g., DARTS) can also be deteriorated by noises inside the graphs, leading to suboptimal architectures.

To solve the problem, we propose **G**raph differentiable **A**rchitecture **S**earch model with **S**tructure **O**ptimization (GASSO), which allows differentiable search of the architecture with gradient descent and is capable of searching the optimal architecture as well as adjusting graph structure adaptively through a joint optimization scheme. The graph structure adjustment in our proposed GASSO model serves as an adaptive noise reduction to enhance the architecture search performance. The framework of GASSO is shown in Figure 1. We employ differentiable graph structure to allow us to optimize graph structure by gradient based methods during the training process, and we use hidden feature smoothness to evaluate the edge importance and constrain edge weights. Overall, we jointly optimize the parameters of neural architecture and graph structure in an iterative updating manner. We evaluate our model on several widely used graph benchmark datasets including CiteSeer, Cora and PubMed. The experimental results show that our proposed GASSO model can outperform state-of-the-art methods and demonstrate a better denoising ability than existing graph architecture search model using DARTS and other GNNs models. To summarize, this paper makes the following contributions: (1) We theoretically explore how graph neural architecture search is able to select the desired GNN architectures, to the best of knowledge, for the first time, showing that i) gradient based graph

architecture search prefers operations who can help to correct the predictions on hard data, and ii) different operations fit graphs with different amount of information in the node features and graph structures. (2) We design measurement study with experiments to show that i) gradient based graph architecture search is able to select operations based on the usefulness of the information in graphs, and ii) noises in graph features and structures can deteriorate the architecture search performance. (3)We propose **G**raph differentiable **A**rchitecture **S**earch model with **S**tructure **O**ptimization (GASSO), which searches the optimal architecture as well as adjusts graph structure adaptively through a joint optimization scheme. Experimental results on several graph benchmark datasets demonstrate the superiority of our GASSO model against state-of-the-art methods.

## 2   Related Work

### 2.1   Graph Neural Network

As an effective framework for graph representation learning, GNN [1–3, 9–13] follows a neighborhood aggregation scheme. At each iteration, representations of nodes are generated through aggregating their neighbors' representations. For instance, graph convolutional neural network (GCN) [1] takes the average of all neighbor nodes' features as aggregation. Different from GCN, Graph Attention Network (GAT) [2] treats neighbors unequally. It calculates the attention on each neighbor. Therefore, the aggregation can emphasize representations of more important neighbor nodes with larger attentions. The vectorized representation of a given node in the graph after $k$ iterations can capture both structural (topological) and semantic (attributed) information within the region of the target node's $k$-hop neighborhood. As such, GNN characterizes the whole graph by aggregating node representations [1, 14], making themselves achieve the state-of-the-art results for tasks including node and graph classifications. Like other types of deep neural networks, GNNs are vulnerable to noise and adversarial attacks. Graph attackers usually adopt data perturbation by changing the node features or graph structure [15, 16]. How to denoise and defend graph adversarial attacks is also a popular question [17–19]. Graph structure learning is presented in recent GNN methods [20–22]. Since graph structure plays an important role in GNN scheme, certain noise in the original graph is extremely harmful to GNNs. Graph structure learning aims to generate a clean graph during the training procedure. To achieve this, prior constraints such as low rank and smoothness are utilized. Learning a new graph structure improves the model's ability of denoising, as well as its robustness to adversarial attacks of GNNs.

### 2.2   (Graph) Neural Architecture Search

Recent years have witnessed a significant surge in research on automated machine learning, including hyper-parameter optimization [23–26] and Neural Architecture Search (NAS) methods [6–8, 27–30] aim at designing an neural architecture automatically for certain tasks. Since the architecture search space is discrete, reinforce learning [6, 7] and evolution algorithm [31, 32] are often used in NAS methods. Besides, transferring the discrete architecture search space into a differentiable space is another strategy for solving the NAS problems. DARTS [8] and SNAS [33] construct a super network where each operation is mixed by all candidate operations, making it possible to update the architecture as well as the weights simultaneously through the classical gradient descent method. NAS has also been applied to graph tasks, achieving competitive performance with the state-of-the-art performances. Existing works [4, 5, 34–36] mainly focus on defining a proper search space and search strategy for GNNs. However, the mechanism of NAS methods to select architectures and the relationship between NAS methods and GNN denoising ability have not been noticed.

## 3   Exploring Architecture Search for Graph

### 3.1   Preliminaries: Differentiable Architecture Search

We use DARTS [8], a representative NAS method with both conciseness and effectiveness as the exploration object in this paper. In DARTS, neural architecture is represented as a directed acyclic graph (DAG) where the input data is the source node, the output is the sink node, and the edges represent operations (e.g. a GCN layer, or a linear layer) adopted on the data. Following a micro

search space setting, DARTS only searches which operation should be selected and how the nodes in the DAG should be connected. The calculation inside each operation is predefined.

There are two phases in DARTS procedure, the searching phase and the evaluation phase. In searching phase, a super network (shown in Figure 1) is constructed, where edges exist between each two nodes, i.e., $e_{i,j}$ exists, $\forall 0 \leq i < j \leq N-1$. Each edge is a mixed operation that can be calculated by $e_{i,j}(\mathbf{x}_i) = \sum_{o \in \mathcal{O}} \frac{exp\{\alpha_{i,j}^o\}}{\sum_{o' \in \mathbf{o}} exp\{\alpha_{i,j}^{o'}\}} \cdot o(\mathbf{x}_i)$, where $\mathbf{x_i}$ is the output of node $i$, $\mathcal{O}$ is candidate operation set, $\alpha$ is learnable architecture parameters. After calculating the mixed weighted sum, each node aggregates all input edges by $\mathbf{x}_j = \sum_{i<j} e_{i,j}(\mathbf{x}_i)$. As such, all possible operations at all possible positions are contained in the super network. DARTS uses gradient based methods to optimize both architecture parameters $\alpha$ and operation parameters by a bi-level optimization scheme. Since the parameters are all trained in the super network, $W$ of different architectures are shared with each other, the mathematical formulation is as follows:

$$\min_{\mathcal{A}} \quad \mathcal{L}_{val}(W^*, \mathcal{A}) \qquad s.t. \quad W^* = argmin_W \mathbb{E}_{\mathcal{A} \in \Gamma(\mathcal{A})} \mathcal{L}_{train}(W, \mathcal{A}). \tag{1}$$

Here, $\Gamma(\mathcal{A})$ is architecture distribution learned in the searching phase. In this formulation, $W$ and $\mathcal{A}$ are independently learned, improving the search efficiency.

At the end of the training procedure, the operation with the maximum $\alpha$ of each node is chosen, and the selected operations compose the optimal architecture. Then comes to evaluation phase, a new network based on the designed architecture is constructed. This network will be trained from scratch and finally get tested after retraining.

## 3.2 Theoretical Analysis of Architecture Parameters

We next explore the DARTS behavior on graph task, aiming to answer (i) *how does graph neural architecture search select its desired architectures?* and (ii) *how optimal are the architectures selected by graph neural architecture search?* We firstly provide theoretically analysis on (i).

Mixed operation is the key part in DARTS, where the architecture parameters controls the operation selection. However, there is no theoretical analysis on how these architecture parameters change according to the performance of different operations. To explore how these architecture parameters change during training procedure, we firstly analyze a simple scene of a binary classification problem, and there is only one layer with two candidate operations needs searching in the super network. The mixed operation is $F(\mathbf{x}_k) = o_1 f_1(\mathbf{x}_k) + o_2 f_2(\mathbf{x}_k)$. Here, $o_1 = \frac{exp\{\alpha_1\}}{exp\{\alpha_1\} + exp\{\alpha_2\}}$ and $o_2 = \frac{exp\{\alpha_2\}}{exp\{\alpha_1\} + exp\{\alpha_2\}}$. $f_1$ and $f_2$ are candidate operations. We use gradient methods to minimize binary cross entropy loss combined with a sigmoid function: $H(F) = -\sum_k [y_k \cdot ln\sigma(F(\mathbf{x}_k)) + (1 - y_k) \cdot ln(1 - \sigma(F(\mathbf{x}_k)))]$. Then we can conduct following theorem with proof in Appendix A.1.

**Theorem 1** *The changes of operation weights $o_n$ after one step of gradient descent w.r.t. architecture parameters depend on the score $\sum_k [w_k f_n(\mathbf{x}_k)]$, where $w_k = \sigma(F(\mathbf{x}_k)) - y_k$. Specifically, if $\sum_k [w_k f_1(\mathbf{x}_k)] < \sum_k [w_k f_2(\mathbf{x}_k)]$, then $o_1$ increases and $o_2$ decreases, and vice versa.*

Theorem 1 tells us that DARTS judges different candidate operations by a score, which can be seen as a weighted sum of logits. As for the weights, we have $w_k \in (-1, 1)$, whether it is positive or negative depends on the data label. If $y_k = 0$, the weight is a positive number $\sigma(F(\mathbf{x}_k))$, which means those data with bad prediction given by the super network have larger weights. If only considering those scores with positive weights, the candidate operation giving better prediction has smaller logits. Summing them up will get smaller score and increase architecture weight $o_n$. We can reach a similar conclusion if we analyze data with label $y_k = 1$. In a nutshell, we know DARTS prefers those operations which can help to correct the predictions over hard data.

## 3.3 Synthetic Graph Experiment Setting

We further construct a synthetic graph which is composed of 600 nodes with 10 types, 60 nodes per type. The node features of a certain type follow a Gaussian distribution $\mathcal{N}(\mu_i, \sigma_1)$ (to be simple, we write one-dimension Gaussian distribution here, all dimensions are the same and independent, the same below), where $\mu_i$ is sampled from another Gaussian distribution $\mathcal{N}(0, \sigma_2)$. $\sigma_1$ and $\sigma_2$ are predefined parameters. Here we can use $\beta = \frac{\sigma_2}{\sigma_1}$ to represent the difficulty of classifying nodes by

their features. If $\beta$ is large, intra-group node features are close while inter-group node features are far away ("group" denotes node label here), thus it is easy to classify nodes by their features, vice versa. This difficulty also reflexes the ratio of noises and type-relevant information inside the features.

As for the edges, we also set two parameters $p_1$ and $p_2$. $p_1$ is the probability of generating an edge between two intra-group nodes while $p_2$ is that of two inter-group nodes. We define $\delta = \frac{p_1}{p_2}$, which can reflex the difficulty of classifying nodes by message passing scheme. If $\delta$ is large, intra-group nodes appear more often in the neighborhood, making it easier to classify nodes by aggregating neighborhood information, and vice versa. This difficulty also reflexes the ratio of noises and type-relevant information inside the graph structure.

We generate synthetic graphs with different $\beta$ and $\delta$, then split the nodes in supervised setting for node classification task. To theoretically explore how different operations behave under different settings, we analyze a simple scene here. We only consider one target node, which is connected with only two classes of nodes, one is the same as its own class. The node feature channel is 1. The candidate operations only contain *linear* and *GCN* [1], they can represent two types of operations, message-passing operations and those without message passing.

**Theorem 2** *Under our synthetic graph setting, let $n$ be the number of edges connected the target node, the relative distance between the centers of two classes is $|D|$, which follows $D \sim \mathcal{N}(0, \beta^2)$. Then, the probability of that linear operation gives more accurate prediction than GCN on the target node is $P = \Phi[\frac{\sqrt{2}n|D|}{(\delta+1)\sqrt{(n+1)(n+2)}}]$.*

The proof of Theorem 2 is in Appendix A.2. Theorem 2 shows that the information gained by different operations are influenced by $\beta$, $\delta$ and $n$. When $\beta$ increases, the expectation of $|D|$ increases, thus the probability to choose *linear* increases. When $\delta$ increases, the probability to choose *GCN* increases. the number of edges can influence the probability as well, but weaker than $\beta$ and $\delta$.

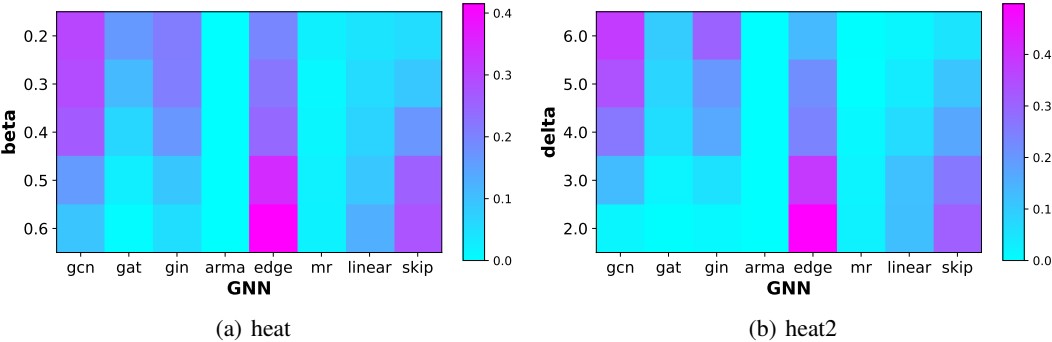

(a) heat            (b) heat2

Figure 2: The frequency of GNNs appeared in optimal architecture in different graph setting

### 3.4 Operation Selection

To discover how $\beta$ and $\delta$ influence DARTS in practice, we generate synthetic graphs by controlling one variable when we change the other one. We set $\beta = 0.4$ while changing $\delta$, and set $\delta = 4.0$ while changing $\beta$. For each setting of $\beta$ and $\delta$, we generate 100 different graphs and adopt DARTS on it. We choose *GCN* [1], *GAT* [2], *GIN* [3], *MRConv* [37], *EdgeConv* [38], *linear*, *skip connect* and *zero* operation as the candidate operations in DARTS. We set the number of operations to be 6. Since GNNs usually are very shallow, we mainly explore DARTS behavior on operation selection but not the connections. We count the times of each operation that appears in the optimal architecture and show the frequency in Figure 2.

According to the results above, we found that for different graphs, DARTS suggests using different operations. To be more specific, for graphs with more structure information (large $\delta$) less feature information (small $\beta$), DARTS tends to select typical message passing operations, such as GIN. When there is less structure information (small $\delta$) and more feature information (large $\beta$), DARTS is more likely to select operations like *linear* and *skip connect* to prevent message passing. The results agree with Theorem 2.

Table 1: The GNN performances under different settings of the searching phase and evaluation phase. "S" indicates the setting of searching phase. "E" indicates the setting of evaluation phase. All the results are average of 100 runs, the variances of each result are listed in Appendix C.1. **Left**: under different $\beta$. **Right**: under different $\delta$.

| E \ S | 0.2 | 0.3 | 0.4 | 0.5 | 0.6 |
|---|---|---|---|---|---|
| $\beta = 0.2$ | **56.4** | 70.9 | 82.0 | 87.9 | 91.6 |
| $\beta = 0.3$ | 55.5 | **73.1** | 85.1 | 91.7 | 94.4 |
| $\beta = 0.4$ | 52.3 | 73.1 | 86.5 | 93.0 | 96.3 |
| $\beta = 0.5$ | 48.5 | 72.7 | **88.6** | 96.1 | 98.6 |
| $\beta = 0.6$ | 43.0 | 69.4 | 88.1 | **96.3** | **99.0** |

| E \ S | 6.0 | 5.0 | 4.0 | 3.0 | 2.0 |
|---|---|---|---|---|---|
| $\delta = 6.0$ | 93.4 | 88.4 | 80.6 | 68.9 | 55.1 |
| $\delta = 5.0$ | 94.2 | 90.1 | 83.6 | 79.2 | 62.9 |
| $\delta = 4.0$ | **94.2** | **91.3** | 86.3 | 79.2 | 70.5 |
| $\delta = 3.0$ | 92.9 | 91.0 | **88.7** | **85.8** | 82.1 |
| $\delta = 2.0$ | 86.8 | 86.3 | 85.8 | 85.4 | **84.8** |

Here we can give some explanations for (i) by combining two theorems and the experiment results. A GNN can be regarded as an information extractor, which collects useful messages inside the graph to give accurate node classification. The operations in the search space have different ability to extract information from node feature and graph structure, e.g., *GCN* has the ability to extract information from graph structure but reduce the influence of the node's own feature, while *linear* only uses the feature information and ignores graph structure. DARTS aims to find the optimal information extractor to help correct the prediction of the super network. To achieve this, it should judge how the node feature and graph structure is useful for the task. If the graph structure is useful, DARTS will take advantage of it and select operations that can extract information from the topology. If the node features are useful, DARTS will decide to prevent message passing to make full use of the features.

In addition, we can find that the heat map of the two experiment results are very similar, revealing that DARTS judges the relative amount of the two types of information but not the absolute amount. This also agrees with Theorem 2, e.g., as long as node features are more useful than graph structure, the structure is thought to have certain noise that is harmful to the task, so message passing is preferred to be prevented.

*On denoising issue.* We have concluded that DARTS has the ability to judge which part inside the graph is useful and select proper operations. This behavior motivates us to test the denoising ability of DARTS. When random noise perturbs a clean graph, DARTS can rejudge the usefulness of these two parts and give new architecture suggestions according to the current condition. Thus DARTS may have denoising ability. We make experiments on real graph datasets in Section 5. The results show its denoising ability is unstable. Therefore, we move on to explore how accurate DARTS selects operations.

### 3.5 Accuracy of Operation Selection

To further explore the answer to (ii), we save the architectures searched on graphs with each $\beta$ and $\delta$ in Figure 2, and transfer them on graphs with different settings to do the evaluation. We also control one variable while changing the other one. The results are shown in Table 1.

Following normal sense, the architecture searched on a graph should perform well on the same graph, i.e., the elements on the main diagonal in the tables should be the maximum elements of theirs columns. However, we find that when evaluating on a certain setting, architecture searched from another setting can perform well, even better than the architecture searched at the same setting. E.g., the best architecture evaluated on the graph with $\delta = 5$ is searched from the graph with $\delta = 4$. This phenomenon can answer (ii). DARTS may not be able to accurately select the optimal architecture with the original graph.

We can also give our explanation here. Although DARTS can judge the usefulness of different information, they cannot get an accurate answer, e.g., when $\delta = 6$, graph structure contains useful classification information. However, DARTS may have overconfidence in it and make radical decisions, aggregating too much message from the neighborhood which includes noises and cause over-fitting. Less graph structure information ($\delta = 4, 5$) lets DARTS become more conservative about aggregation. Architectures searched at these settings perform better when evaluating at $\delta = 6$ setting. Similarly, DARTS may misjudge the information's usefulness in other settings.

Based on the above exploration, we have found that DARTS cannot design the optimal architecture due to certain noise in the original graph. Message passing scheme used in GNNs smooths node

features, which can also be seen as a denoising process. Therefore, denoising graph structure is what we should consider more. Combining architecture search and structure learning becomes a natural way. In the next section, we will introduce our method which jointly optimizes neural architecture and graph structure.

## 4 Joint optimization of graph architecture and structure

### 4.1 Differentiable Graph Structure

In most real graph datasets, the graph structure is discrete, i.e., the edges given in the graph are unweighted. Although these edges contain crucial information, certain noise is brought into the data as well. Take citation dataset for example, a paper cites other papers for many different reasons. Some citations are helpful for paper classification, such as citing related works and baselines. Others may have less use, such as citing a mathematical method. However, the graph treats all these different types of citation equally. The classification model may be confused by those unhelpful edges. This phenomenon can be also found in other types of graphs. In addition, the discrete space of graph structure makes it difficult to optimize the structure. Thus, it is necessary to differentiate the edges in the graph.

We apply differentiable graph structure here. We use a parameter to represent the weight of each edge. To restrict the weight is in $(0, 1)$ during message passing, we use a sigmoid function to the parameter. Thus, the parameter matrix used in message passing can be represented as $G_p = normalize(sigmoid(G))$, where $G$ is the parameter. The $normalize$ function includes adding self-loops, i.e, we can only change the weight of normal edges, the weight of self-loops is fixed. In the message passing procedure, edge weights are used as multipliers before aggregation.

We benefit from the differentiable graph structure in two ways. One is we can distinguish the helpfulness of edges, and aggregate more useful information during message passing. The other one is we can use gradient based methods to optimize the structure. We should note this differentiable graph structure fits most of GNNs by just replacing the adjacent matrix with the parameter matrix $G_p$.

### 4.2 Learning by Hidden Feature Smoothness Constraint

An important assumption in graph is the first-order proximity. It is proved in [21, 22] that more links between intra-group nodes and fewer links between inter-group nodes improve the node classification accuracy. However, we cannot get all node labels in the training phase. We use hidden feature $H$ to approximate the node labels, which is the matrix indicating the probability of which type the nodes belong to. It is proper to use $H$ to represent the node label. To minimize the connections between inter-group nodes, we can use the hidden feature smoothness as a regularizer. In condition that the model has learnt some knowledge, the model will give good predictions about node labels. Thus, the edge weights whose connected nodes have similar hidden features are larger than those have different hidden features. By updating the structure, similar node features are aggregated more in later GNN procedure, improving the denoising ability of the algorithm. However, if we only adopt this loss to optimize the structure. All edge weight will be reduced to $0$ at convergence. Thus we combine the smoothness loss with a distance loss, which can be formulated as:

$$\mathcal{L}_s = \lambda \sum_{i,j}^{N} G_{ij} \parallel \mathbf{h}_i - \mathbf{h}_j \parallel_2 + \sum_{i,j}^{N} (G_{ij} - G_{o,ij})^2, \tag{2}$$

where $G_{ij}$ is the weight parameter of the edge between node $i$ and $j$, $G_{o,ij}$ is the elements in the initialization of $G$, we initial all edge weights as $0.5$, $\lambda$ is a hyper-parameter to control the contribution of the hidden feature smoothness. Hence, the weight of edges between inter-group nodes is reduced. This loss guarantees the first-order proximity to a certain degree. In addition, this loss restricts $G$ not to change too much from the original graph. To save calculation resources, we only learn the weight of edges existing in the original graph in this procedure. The weight of other edges is always kept as $0$. Therefore, the memory and time complexity of our methods is $O(|E|)$. Besides, since $G$ also takes part in the calculation of $\mathcal{L}_s$, we detach $H$ from the computational graph while calculating the gradient of $\mathcal{L}_s$ to $G$ in order to prevent a complex back-propagation.

### 4.3 Training Procedure

We formulate the joint optimization problem as follows:

$$\min_{\mathcal{A}} \mathcal{L}_{val}(W^*, \mathcal{A}, G^*)$$
$$s.t. \quad G^* = argmin_G \mathcal{L}_s(W^*, \mathcal{A}, G),$$
$$W^* = argmin_W \mathbb{E}_{\mathcal{A} \in \Gamma(\mathcal{A})} \mathcal{L}_{train}(W, \mathcal{A}, G). \tag{3}$$

In this problem, there are three parameters $W, \mathcal{A}, G$ needing optimized. It is hard to optimize all of them at the same time. Thus we apply gradient descent methods to these parameters in turns. In the training procedure, we set the first several epochs as warming up, when only $W$ is updated. Since at the very beginning of training, $W$ is unstable and may lead to incorrect hidden features. Using those hidden features to update the other two parameters aggravates the model's unstableness.

Different from DARTS for image tasks where a small proxy model is usually used in the searching phase due to the memory limitation, and a larger network is retrained in the evaluation phase. Here, the number of layers in the super network is the same as a complete GNN. We can directly use the super network to do the evaluation. Interestingly, we find that the super network performs well in node classification tasks, even better than retraining the designed architecture at most times. This phenomenon may be due to the regularization caused by weight sharing, which is one type of ensemble model [39]. Other ensemble models such as dropout [40] and multiple heads [41] have proved their success in deep neural networks. All these models can be regarded as an ensemble of different submodels, which learn different parts of knowledge from the data and provide different perspectives of it. These submodels restrict each other during the training phase, which is de facto a type of regularization, helping reduce over-fitting and improve performance.

## 5 Experiments

### 5.1 Performance Evaluation

| Dataset | Cora | Citeseer | Pubmed |
|---|---|---|---|
| GCN[†] | 87.40 | 79.20 | 88.40 |
| GAT[†] | $87.26 \pm 0.08$ | $77.82 \pm 0.11$ | $86.83 \pm 0.11$ |
| ARMA[†] | $86.06 \pm 0.05$ | $76.50 \pm 0.00$ | $88.70 \pm 0.24$ |
| DropEdge[†] | $87.60 \pm 0.05$ | $78.57 \pm 0.00$ | $87.34 \pm 0.24$ |
| DARTS | $86.18 \pm 0.36$ | $74.96 \pm 0.10$ | $88.38 \pm 0.18$ |
| GDAS | $85.48 \pm 0.30$ | $74.20 \pm 0.11$ | $89.50 \pm 0.14$ |
| ASAP | $85.21 \pm 0.13$ | $75.14 \pm 0.09$ | $88.65 \pm 0.10$ |
| XNAS | $86.80 \pm 0.14$ | $76.33 \pm 0.09$ | $88.61 \pm 0.25$ |
| GraphNAS[‡] | $86.83 \pm 0.56$ | $79.05 \pm 0.28$ | $89.99 \pm 0.43$ |
| GASSO | $\mathbf{87.63 \pm 0.29}$ | $\mathbf{79.61 \pm 0.32}$ | $\mathbf{90.52 \pm 0.24}$ |
| GASSO-V1 | $87.19 \pm 0.24$ | $78.71 \pm 0.16$ | $90.30 \pm 0.46$ |
| GASSO-V2 | $87.17 \pm 0.26$ | $78.62 \pm 0.22$ | $90.13 \pm 0.33$ |
| GASSO-V3 | $87.35 \pm 0.19$ | $78.80 \pm 0.33$ | $90.26 \pm 0.44$ |
| GASSO-V4 | $87.31 \pm 0.21$ | $79.04 \pm 0.34$ | $89.12 \pm 0.18$ |
| GASSO-V5 | $86.26 \pm 0.50$ | $77.08 \pm 0.55$ | $89.34 \pm 0.20$ |
| GASSO-V6[†] | $86.78 \pm 0.24$ | $79.40 \pm 0.35$ | $89.78 \pm 0.20$ |
| GASSO-V7[†] | $87.35 \pm 0.46$ | $78.38 \pm 0.31$ | $89.52 \pm 0.20$ |
| GASSO-V8 | $87.14 \pm 0.37$ | $78.38 \pm 0.24$ | $90.40 \pm 0.39$ |
| GASSO-V9 | $87.37 \pm 0.24$ | $78.22 \pm 0.42$ | $90.24 \pm 0.30$ |

Table 2: Node Classification Performance

† To be fairly compared, we expand the channels of these model to make the number of model parameters comparable with GASSO.
‡ We rerun GraphNAS because their code leaks test set of data.

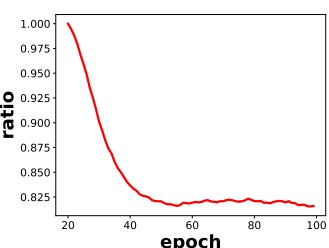

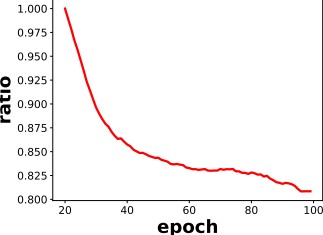

Figure 3: Weight ratio of noisy edges to orignical edges. **Top**: Cora. **Bottom**: Citeseer.

Since structure learning is directly performed on the original graph, we focus on transductive node classification task. We evaluate our model on three widely used citation benchmark datasets, Cora, Citeseer, and Pubmed, where nodes denote papers and edges denote citation relationship. We apply

the full-supervised training fashion used in [42–44]. Hyper-parameter settings are listed in Appendix B.2. We use GCN [1], GAT [2], ARMA [45], DropEdge [42], DARTS [8], GDAS [46], ASAP [47], XNAS [48] and GraphNAS [5] as our baselines, and the hyper-parameter of them are following the original implementation. For our GASSO model, we set the number of layers as 2 in Cora and CiteSeer, 4 in PubMed. The candidate operations contain *GCN* [1], *GAT* [2], *GIN* [3], *MRConv* [37] and *linear*. In the super network, we adopt dropout (p=0.8) before each layer and a ReLU function after each layer. We summarize node classification accuracy in Table 2. All results in the table are averaged over 10 runs with random parameter initialization, and the standard deviations are also provided in the table.

We can find that our model GASSO outperforms all baselines on all three benchmark datasets. The results show that our proposed model has the power to effectively utilize information in the original graph, giving better node label predictions.

We conduct further experiments on three larger graph benchmarks: Physics, CoraFull and ogbn-arxiv. For Physics and CoraFull, we randomly split train/valid/test set as 50:25:25. For ogbn-arxiv, we follow the default setting. We narrow down the operation search space to *GCN*, *GAT* and *linear* because of the memory limit. The results shown in Table 3 indicating that our model still performs well in larger graphs.

| Dataset | Physics | CoraFull | ogbn-arxiv |
|---------|---------|----------|------------|
| GCN | 95.94 | 68.08 | 70.39 |
| GAT | 95.86 | 65.78 | 68.53 |
| DARTS | 95.74 | 68.51 | 69.52 |
| GASSO | 96.38 | 68.89 | 70.52 |

Table 3: Node Classification Performance

## 5.2 Ablation Study

In our scheme, the most two important parts are neural architecture search and graph structure learning. To further evaluate the contribution of these two components and their cooperation, we conduct an ablation study on them. We design several variant models, which can be divided into three parts. 1) We change the graph structure learning model in the first part of variants. For GASSO-V1, we remove the structure learning part, i.e., the model becomes pure DARTS with our initial $G$. For GASSO-V2, we replace hidden feature smoothness with original feature smoothness, which is used in [20]. For GASSO-V3: we replace hidden feature smoothness with one-hot predicted label smoothness, which is similar with [21]. For GASSO-V4: we directly calculate edge weight based on similarities between nodes. 2) In the second part, we change the neural architecture search method. For GASSO-V5, we do not update the neural architecture, but the super network is still kept, i.e., all candidate operations have equal weights from the beginning to the end. For GASSO-V6, we fix GCN as the architecture. For GASSO-V7, we fix GAT as the architecture. 3) In the third part, we change the optimization scheme. The aim of setting this part of variants is to prove the cooperation of graph structure learning and neural architecture search. For GASSO-V8: we split the training of neural architecture and graph structure. In the first half of epochs, we only train the graph structure. In the second half, we only train the neural architecture. To be fair, the number of training epochs is doubled, and the warming up procedure is still kept. For GASSO-V9: the same as GASSO-V8, but the order of training neural architecture and graph structure is exchanged.

We examine all these variant on the same datasets, the results are also presented in Table 2. Overall, our model outperforms all of the variants in all three datasets, showing training of both graph architecture and neural architecture contributes to our model.

For part 1), we find that GASSO-V1 also achieves comparable performance with GCN and GAT, showing the great power of ensemble in the super network. But the margin below GASSO proves that graph structure learning is necessary. Interestingly, the performance of GASSO-V2 is not even good as GASSO-V1. It is possible that using original feature smoothness to constrain edges causes more severe over-fitting. The performance of GASSO-V3 indicates that one-hot predicted label smoothness is of some usefulness. But neither the original feature nor the one-hot predicted label is better than hidden feature for graph structure learning. GASSO-V4 is a radical method, which makes the edge weights different at the early stage of training. For part 2), we observe that the performance has an obvious drop in GASSO-V5, demonstrating that the neural architecture training is of significant importance. GASSO-V6 and GASSO-V7 behave differently in three datasets compared with GCN and GAT, showing these datasets have different properties. For part 3), we dicover that GASSO-V8 can be regarded as GASSO-V4 followed by neural architecture search, and GASSO-V9

can be regarded as GASSO-V1 followed by graph structure learning. The margin of GASSO-V8 over GASSO-V4 and margin of GASSO-V9 over GASSO-V1 prove that adding neural architecture search behind graph structure learning can achieve better performances, while adding graph structure learning behind neural architecture search causes unstable influence. Nevertheless, neither GASSO-V8 nor GASSO-V9 outperforms GASSO, demonstrating that the cooperation training in our scheme play a key role in optimizing the model.

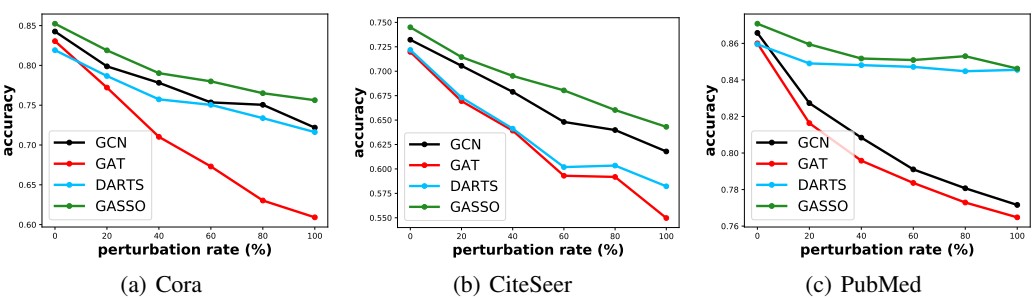

| (a) Cora | (b) CiteSeer | (c) PubMed |

Figure 4: Accuracy with noisy edges

## 5.3  Denoising Analysis

Here we examine the denoising ability of our model. We select the experiment setting following [15, 16, 20], where only the largest connected component is considered, and the split of training/validation/testing set is $10\%/10\%/80\%$. We manually include random edges in the three datasets with different perturbation rates, from $0\%$ to $100\%$ with a step of $20\%$. We compare our methods with baselines GCN, GAT and DARTS. The results are shown in Figure 4. We observe that DARTS has an unstable denoising ability. It performs very well on PubMed, the accuracy almost keeps as it on the original graph even when there is $100\%$ perturbation rate of edges. However, it performs badly in the other two cases, even worse than GCN. Our proposed methods consistently outperform the three baselines in all three datasets. Comparing our proposed GASSO with GCN and GAT, the performance gain gets larger when more noise is added to the graph in all cases. Moreover, GASSO leads a margin of around $6\%$ in CiteSeer and $4\%$ in Cora upon DARTS at $100\%$ perturbation rate setting. To better demonstrate the effectiveness of our denoising process, we show the weight ratio of noisy edges to original edges in Figure 3. We choose the case of $100\%$ perturbation rate in Cora and CiteSeer. We use 20 epochs to warm up. Then the weight ratio begins to decrease. It tends to be stable at around $82.5\%$ in both cases, indicating we can exclude $17.5\%$ of noise effect. All the above results show that GASSO has a great power to reduce noise disturbing NAS methods.

## 6  Conclusion

In this paper, we study how graph architecture search can select desired architectures by conducting measurement study and proposing a Graph Differentianle Architecture Search model with Structure Optimization (GASSO). We find that gradient based graph architecture search can select proper operations according to the usefulness of information inside the graph, and may suffer from noises in graph features and structures. Based on our findings, we proposed our GASSO model, which jointly optimizes graph architecture and structure. Extensive experiments on real graph benchmark datasets demonstrate that the joint optimization of neural architecture and graph structure not only benefits in graph learning but also enhances the denoising ability.

## Acknowledgments and Disclosure of Funding

This work is supported by the National Key Research and Development Program of China No. 2020AAA0106300 and National Natural Science Foundation of China (No. 62050110, No. 62102222).

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
