# Graph Differentiable Architecture Search with Structure Learning

**Yijian Qin, Xin Wang,**\* **Zeyang Zhang, Wenwu Zhu**\*
Tsinghua University
qinyj19@mails.tsinghua.edu.cn, xin_wang@tsinghua.edu.cn
zhangzey@tsinghua.org.cn, wwzhu@tsinghua.edu.cn

## A  Proofs

### A.1  Proof of Theorem 1

**Theorem 1** *The changes of operation weights $o_n$ after one step of gradient descent w.r.t. architecture parameters depend on the score $\sum_k[w_k f_n(\mathbf{x}_k)]$, where $w_k = \sigma(F(\mathbf{x}_k)) - y_k$. Specifically, if $\sum_k[w_k f_1(\mathbf{x}_k)] < \sum_k[w_k f_2(\mathbf{x}_k)]$, then $o_1$ increases and $o_2$ decreases, and vice versa.*

**Proof A.1** *We firstly give Lemma 1:*

**Lemma 1** *The operation weights are caculated by a softmax function. E.g. $o_1 = \frac{exp\{\alpha_1\}}{exp\{\alpha_1\}+exp\{\alpha_2\}}$. Let $g_1$ and $g_2$ are the gradients w.r.t. $\alpha_1$ and $\alpha_2$. The operation weights after a gradient descent step are $o'_1$ and $o'_2$, e.g., $o'_1 = \frac{exp\{\alpha_1-g_1\}}{exp\{\alpha_1-g_1\}+exp\{\alpha_2-g_2\}}$. If $g_1 < g2$, then $o'_1 > o_1$ and $o'_2 < o_2$, and vice versa.*

*Proof of Lemma 1:*

*If $g_1 < g_2$, we should improve $o'_1 > o_1$, that is*

$$
\begin{aligned}
\frac{exp\{\alpha_1 - g_1\}}{exp\{\alpha_1 - g_1\} + exp\{\alpha_2 - g_2\}} &> \frac{exp\{\alpha_1\}}{exp\{\alpha_1\} + exp\{\alpha_2\}} \\
exp\{\alpha_1 - g_1\}(exp\{\alpha_1\} + exp\{\alpha_2\}) &> exp\{\alpha_1\}(exp\{\alpha_1 - g_1\} + exp\{\alpha_2 - g_2\}) \\
exp\{2\alpha_1 - g_1\} + exp\{\alpha_1 + \alpha_2 - g_1\} &> exp\{2\alpha_1 - g_1\} + exp\{\alpha_1 + \alpha_2 - g_2\} \\
exp\{\alpha_1 + \alpha_2 - g_1\} &> exp\{\alpha_1 + \alpha_2 - g_2\} \\
\alpha_1 + \alpha_2 - g_1 &> \alpha_1 + \alpha_2 - g_2 \\
g_1 &< g_2
\end{aligned}
\tag{1}
$$

*We already have $g_1 < g_2$, thus $o'_1 > o_1$. Since $o_1 + o_2 = o'_1 + o'_2 = 1$, we have $o'_2 < o_2$. Lemma 1 is proved.*

*Then we prove Theorem 1 with Lemma 1:*

---

\*Corresponding Authors.

35th Conference on Neural Information Processing Systems (NeurIPS 2021), virtual.

*The binary cross entropy loss is*

$$H(F) = -\sum_k [y_k \cdot ln\sigma(-F(\mathbf{x}_k)) + (1 - y_k) \cdot ln(1 - \sigma(-F(\mathbf{x}_k)))]$$

$$= -\sum_k [y_k \cdot (-ln(1 + exp\{-F(\mathbf{x}_k)\})) + (1 - y_k) \cdot (-F(\mathbf{x}_k) - ln(1 + exp\{-F(\mathbf{x}_k)\}))]$$

$$= \sum_k [ln(1 + exp\{-F(\mathbf{x}_k)\}) + (1 - y_k) \cdot F(\mathbf{x}_k)]$$

$$(2)$$

*Then we have*

$$\frac{\partial H(F)}{\partial o_i} = \sum_k [\frac{-exp\{-F(\mathbf{x}_k)\}}{1 + exp\{-F(\mathbf{x}_k)\}} \cdot f_i(\mathbf{x}_k) + (1 - y_k) \cdot f_i(\mathbf{x}_k)]$$

$$= \sum_k [\sigma(F(\mathbf{x}_k)) - y_k) \cdot f_i(\mathbf{x}_k)] \tag{3}$$

$$= \sum_k [w_k f_i(\mathbf{x}_k)]$$

*Besides, since $o_1 = \frac{exp\{\alpha_1\}}{exp\{\alpha_1\} + exp\{\alpha_2\}}$, we have*

$$\frac{\partial o_1}{\partial \alpha_1} = \frac{exp\{\alpha_1\}(exp\{\alpha_1\} + exp\{\alpha_2\}) - exp\{\alpha_1\} \cdot exp\{\alpha_1\}}{(exp\{\alpha_1\} + exp\{\alpha_2\})^2} = \frac{exp\{\alpha_1 + \alpha_2\}}{(exp\{\alpha_1\} + exp\{\alpha_2\})^2} \tag{4}$$

*Similarly,*

$$\frac{\partial o_2}{\partial \alpha_2} = \frac{exp\{\alpha_1 + \alpha_2\}}{(exp\{\alpha_1\} + exp\{\alpha_2\})^2} \tag{5}$$

*Thus $\frac{\partial o_1}{\partial \alpha_1} = \frac{\partial o_2}{\partial \alpha_2}$.*

*If we have the relationship between the two scores, e.g.,*

$$\sum_k [w_k f_1(\mathbf{x}_k)] < \sum_k [w_k f_2(\mathbf{x}_k)]$$

$$\frac{\partial H(F)}{\partial o_1} < \frac{\partial H(F)}{\partial o_2} \tag{6}$$

$$\frac{\partial H(F)}{\partial o_1} \frac{\partial o_1}{\partial \alpha_1} < \frac{\partial H(F)}{\partial o_2} \frac{\partial o_2}{\partial \alpha_2}$$

$$\nabla_{\alpha_1} < \nabla_{\alpha_2}$$

*By Lemma 1, $o_1$ increases and $o_2$ decreases.* $\qquad\square$

## A.2 Proof of Theorem 2

**Theorem 2** *Under our synthetic graph setting, let $n$ be the number of edges connected the target node, the relative distance between the centers of two classes is $|D|$, which follows $D \sim \mathcal{N}(0, \beta^2)$. Then, the probability of that linear operation gives more accurate prediction than GCN on the target node is $P = \Phi[\frac{\sqrt{2}n|D|}{(\delta+1)\sqrt{(n+1)(n+2)}}]$.*

**Proof A.2** *Firstly we have Property 1, the proof of Property 1 can be easily found in Probability Theory textbook.*

**Property 1** *If $x_1 \sim \mathcal{N}(\mu_1, \sigma_1^2)$, $x_2 \sim \mathcal{N}(\mu_2, \sigma_2^2)$, then $x_1 + x_2 \sim \mathcal{N}(\mu_1 + \mu_2, \sigma_1^2 + \sigma_2^2)$*

*Since the centers of the two classes follows $\mu_1, \mu_2 \sim \mathcal{N}(0, \sigma_2^2)$, according to Property 1, the absolute distance between these two centers is $d \sim \mathcal{N}(0, 2\sigma_2^2)$. Since $D \sim \mathcal{N}(0, \beta^2)$ and $\beta = \frac{\sigma_2}{\sigma_1}$, we know the relationship between $D$ and $d$ is $\frac{d}{D} = \sqrt{2}\sigma_1$. We assume the center of the target node's class is $\mu$, the center of the other class is $\mu + |d|$. Thus the target node's feature follows $x_t \sim \mathcal{N}(\mu, \sigma_1)$. The node feature of the other class follows $x \sim \mathcal{N}(\mu + |d|, \sigma_1)$.*

*GCN can be seen as a message passing process followed by a linear layer, where the message passing process is average of all node neighbors. The number of target node's intra-group neighbors is $\frac{\delta n}{\delta+1} + 1$ (include self-loop), while the number of target node's inter-group neighbors is $\frac{n}{\delta+1}$. Let the probability of a target node's neighbor is an inter-group neighbor is $p$, where*

$$p = \frac{\frac{n}{\delta+1}}{\frac{\delta n}{\delta+1} + 1 + \frac{n}{\delta+1}} = \frac{n}{(\delta+1)(n+1)}. \tag{7}$$

*Let $x_G$ represent the node feature after message passing process. We have $x_G \sim \mathcal{N}(\mu + p|d|, \frac{\sigma_1^2}{n+1})$*

*Linear operation gives more accurate prediction than GCN on the target node is equivalent to $x_t < x_G$. Let $\Delta = x_t - x_G$, we have $\Delta \sim \mathcal{N}(-p|d|, \frac{n+2}{n+1}\sigma_1^2)$. Thus we have*

$$
\begin{aligned}
P(x_t < x_G) &= P(\Delta < 0) \\
&= \Phi\left(\frac{p \cdot |d|}{\sqrt{\frac{n+2}{n+1}}\sigma_1}\right) \\
&= \Phi\left(\frac{\frac{n}{(\delta+1)(n+1)} \cdot \sqrt{2}\sigma_1|D|}{\sqrt{\frac{n+2}{n+1}}\sigma_1}\right) \\
&= \Phi\left[\frac{\sqrt{2}n|D|}{(\delta+1)\sqrt{(n+1)(n+2)}}\right]
\end{aligned}
\tag{8}
$$

$\square$

# B Algorithm and hyper-parameters

## B.1 Algorithm

---
**Algorithm 1:** GASSO

---
**Input:** Graph features $X$, graph adjacent matrix $A$, learning rate $\eta_W$, $\eta_\mathcal{A}$ and $\eta_G$
**Output:** Prediction of labels $Y'$
1 Initial the super network $f$ with parameter $W$, $\mathcal{A}$;
2 Adopt DropEdge to $A$ and initial the structure parameter $G$;
3 **while** *not converge* **do**
4     $W = W - \eta_W \nabla_W \mathcal{L}(W, \mathcal{A}, G)$;
5     **if** *is warming up* **then**
6        Continue;
7     **end**
8     $G = G - \eta_G \nabla_G \mathcal{L}_s(W, \mathcal{A}, G)$;
9     $\mathcal{A} = \mathcal{A} - \eta_\mathcal{A} \nabla_\mathcal{A} \mathcal{L}(W, \mathcal{A}, G)$;
10 **end**
11 $H = f_{W, \mathcal{A}, G}(X)$;
12 Return $Y'_i = argmax_j H_{ij}$

---

## B.2 Hyper-paramter settings

We initialize structure parameter $G$ as a null matrix, thus the weights of all edges in $G_p$ are $0.5$ after the sigmoid function. We use Adam optimizer for all three parameters $W$, $\mathcal{A}$ and $G$, the learning

Table 1: The GNN performances under different settings of the searching phase and evaluation phase. "S" indicates the setting of searching phase. "E" indicates the setting of evaluation phase. All the results are average of 100 runs, the variances of each result are listed in Appendix C.1. **Left**: under different $\beta$. **Right**: under different $\delta$.

| S \ E | $\beta = 0.2$ | $\beta = 0.3$ | $\beta = 0.4$ | $\beta = 0.5$ | $\beta = 0.6$ |
|---|---|---|---|---|---|
| $\beta = 0.2$ | $\mathbf{56.4 \pm 4.9}$ | $70.9 \pm 4.9$ | $82.0 \pm 5.1$ | $87.9 \pm 5.0$ | $91.6 \pm 4.9$ |
| $\beta = 0.3$ | $55.5 \pm 5.1$ | $\mathbf{73.1 \pm 4.4}$ | $85.1 \pm 5.2$ | $91.7 \pm 5.1$ | $94.4 \pm 4.6$ |
| $\beta = 0.4$ | $52.3 \pm 5.6$ | $73.1 \pm 4.8$ | $86.5 \pm 5.4$ | $93.0 \pm 5.0$ | $96.3 \pm 4.2$ |
| $\beta = 0.5$ | $48.5 \pm 7.1$ | $72.7 \pm 5.5$ | $\mathbf{88.6 \pm 3.1}$ | $96.1 \pm 1.8$ | $98.6 \pm 1.2$ |
| $\beta = 0.6$ | $43.0 \pm 6.4$ | $69.4 \pm 5.8$ | $88.1 \pm 3.2$ | $\mathbf{96.3 \pm 1.6}$ | $\mathbf{99.0 \pm 1.0}$ |

| S \ E | $\delta = 6.0$ | $\delta = 5.0$ | $\delta = 4.0$ | $\delta = 3.0$ | $\delta = 2.0$ |
|---|---|---|---|---|---|
| $\delta = 6.0$ | $93.4 \pm 2.2$ | $88.4 \pm 3.0$ | $80.6 \pm 4.4$ | $68.9 \pm 6.7$ | $55.1 \pm 8.1$ |
| $\delta = 5.0$ | $94.2 \pm 2.1$ | $90.1 \pm 3.5$ | $83.6 \pm 5.5$ | $79.2 \pm 8.4$ | $62.9 \pm 11.5$ |
| $\delta = 4.0$ | $\mathbf{94.2 \pm 2.1}$ | $\mathbf{91.3 \pm 3.0}$ | $86.3 \pm 5.2$ | $79.2 \pm 8.4$ | $70.5 \pm 12.4$ |
| $\delta = 3.0$ | $92.9 \pm 4.1$ | $91.0 \pm 3.3$ | $\mathbf{88.7 \pm 3.3}$ | $\mathbf{85.8 \pm 3.4}$ | $82.1 \pm 5.5$ |
| $\delta = 2.0$ | $86.8 \pm 4.6$ | $86.3 \pm 4.6$ | $85.8 \pm 4.4$ | $85.4 \pm 3.6$ | $\mathbf{84.8 \pm 3.4}$ |

rates are set as $\eta_W = 0.01$, $\eta_{\mathcal{A}} = 0.03$ and $\eta_G = 0.04$, the weight decay for $W$ is $1e^{-4}$, and $0$ for the other two parameters. The DropEdge probability in PubMed is $0.2$, no DropEdge in Cora or CiteSeer. The hyper-parameter $\lambda$ which controls the hidden feature smoothness is set to be $0.125$.

We set the number of layers as $2$ in Cora and CiteSeer, $4$ in PubMed. The candidate operations contain *GCN* [1], *GAT* [2], *GIN* [3], *MRConv* [4] and *linear*. In the super network, we adopt dropout (p=0.6) before each layer and a ReLU function after each layer.

## C  Experiment results

### C.1  The variance of synthetic graph experiment

We show the variance of synthetic graph experiment in Table 1 to endorse our analysis in Section 3. The table shows that the variance of accuracy is relatively big in the experiment setting. However, all the results are average of 100 runs. Considering the variance of average of 100 tests is $\frac{1}{100}$ of variance of a single test, the relationship among the accuracy is credible. Thus our analysis based on the experiment results makes sense.

### C.2  Hyper-parameter Analysis

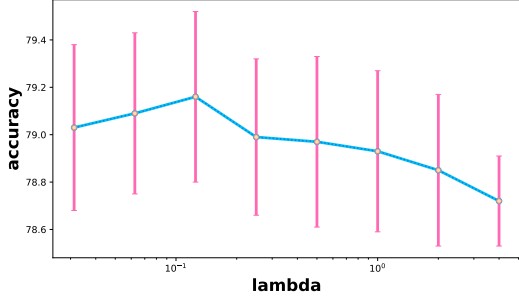

Figure 1: Accuracy on CiteSeer with different lambda

We explore the sensitivity of our hyper-parameter $\lambda$, which is used in Equation (**??**) to control the restriction of hidden feature smoothness. When $\lambda$ gets larger, the weight difference between smooth

edges and non-smooth edges gets larger, and vice versa. If $\lambda$ is set to $0$, the initial $G$ will not be changed.

We vary $\lambda$ from $(0.03125, 4.0)$ in a log scale of base 2. We report the performance changes on CiteSeer in Figure 1. The figure shows that the performance is not very sensitive to $\lambda$, which varies within a $0.5\%$ range in our $\lambda$ space. There is still a peak around $\lambda = 0.125$ where we achieve the best performance.

## D Discussions

In this part, we discuss some key ideas in our methods and their distinction with others.

**with GAT**. In GAT [2], edges are also given weight during the message passing procedure, so nodes can aggregate more information from useful neighbors. GAT gives the weight from the node's perspective. The weight is calculated by the connected node features and at each layer, and asymmetric for the two nodes. It is the node that determines what to aggregate at each convolution layer. Differently, we adopt edge perspective in our design, the edge weight represents the edge's importance in the graph, which is more general. We use the same weight for both connected nodes, for all GNN operations, and for all layers. This perspective of edge weight is more comprehensive.

**with Other Structure Learning Methods**. There are also other graph models containing structure learning, aiming at denoising the original data and enhancing robustness, such as TO-GCN [5] and Pro-GNN [6]. Both of them use first-order proximity to update the graph structure. In TO-GCN, the predicted labels are used to measure node similarity. However, the predicted labels are discrete relaxation of node features, which cannot accurately represent the node. In Pro-GNN, the original features are used to measure node similarity, which makes the structure learning separated from the model training procedure, i.e., its structure learning phase can be easily moved before model training, constructing a two-stage scheme. We use the hidden feature instead, combining structure learning with the training of the model and architecture while keeping accuracy at the same time. Further exploration is presented in Section 5.

**with GraphMix**. GraphMix [7] is a graph regularization model where a fully-connected network (FCN) is used and its parameter is shared with GCN. The FCN is trained by interpolation-based techniques, restricting the training of GCN to reduce over-fitting. Thus, GraphMix can also be seen as ensemble model with regularization. In our model, weights are not shared among different types of operations, but within the same operations at the layer among different architectures. These two types of parameter sharing are orthometric with each other.