# OpenReview forum: "Graph Differentiable Architecture Search with Structure Learning"
_NeurIPS.cc/2021/Conference — NeurIPS 2021 Poster_

### Official Review · Reviewer_fXBu · 2021-07-14

**Rating:** 6
**Confidence:** 4

**Summary:**

This paper combines the differentiable graph structure learning and neural architecture search together. By conducting the two iteratively, the framework can automate GNN design for tasks without given graph. It also provide an analysis that NAS could help balance GNN and MLP by the informativeness of graph against structure.

**Limitations And Societal Impact:**

The main limitation is that the structure learning could not scale to large-scale graph or new nodes in inductive setting.

**Main Review:**

This paper firstly conducts an analysis about which GNN architecture (specifically, which operator) will be learned given a graph by DARTS. They mainly focus on analyzing the search result with respect to the informativeness of graph structure against node feature. The author construct many synthetic graph by stochastic block model, and see how the predicted operator by DARTS is related to the informativeness of structure. The finding is that if a graph is noisy and the structure cannot provide more information than feature, the DARTS will flavor MLP and skip connection against message passing operator.

The analysis of this part is reasonable but not so surprising. Also in real-world the underlying graph structure could be more complicated than the synthetic graph the paper uses. For example, probably only a portion of nodes has noisy edges. Also, it would be interesting to go deeper into the analysis. For example, whether the predicted operator per layer will be different, and how this result is associated with some graph properties. Also, many similar and more in-depth findings have been revealed in "Design Space for Graph Neural Networks". The authors could consider follow some of their experimental setup to strengthen the analysis part.


Given the finding, the author then tries to introduce structure learning to denoise the graph for better NAS. Though the idea is reasonable and there exists several structure learning works, this paper utilizes the most straightforward version: regard the graph as a weighted complete graph and adjust the pairwise weights as edge parameter. Such method cannot scale up to large-scale graph as the edge number goes quadratically to node size, and also it cannot be inductive to new nodes. I highly recommend the authors refer to some recent works on structure learning, including but not limiting to:

[1] Graph Structure Learning for Robust Graph Neural Networks

[2] Neural Relational Inference for Interacting Systems

[3] Learning discrete structures for graph neural networks

......

Finally, the authors only conduct experiments on three small node classification datasets. It would be more convincing to add results on some large-scale graph datasets, such as the ones in OGB, as the scale of network data is very important to the quality of both structure learning and NAS.

**Time Spent Reviewing:**

4

---

> ### Author Response · Authors · 2021-08-10
> **Response to Reviewer fXBu**
>
> Thank you for your reviewing efforts and constructive comments. We address reviewer’s concerns point by point.
>
> *Comment 1: In real-world the underlying graph structure could be more complicated than the synthetic graph the paper uses. For example, probably only a portion of nodes has noisy edges.*
>
> - The aim of using synthetic setting is to explore DARTS behaviors and answer the two questions by convenient simulation control of structure and node feature information in the graph, which can provide us with inspiration to design better Graph NAS algorithms. We follow previous work [1] to conduct the synthetic experiment.
> - The situation that only portion of the nodes have noisy edges in real-world is exactly what inspired us to add structural learning to NAS procedure, thus we can weaken the influence of noisy edges and keep normal edges.
>
> [1] Wang, Xiao, et al. “Am-gcn: Adaptive multi-channel graph convolutional networks.” Proceedings of the 26th ACM SIGKDD International conference on knowledge discovery & data mining. 2020.
>
> *Comment 2: It would be interesting to go deeper into the analysis. For example, whether the predicted operator per layer will be different, and how this result is associated with some graph properties. Also, many similar and more in-depth findings have been revealed in "Design Space for Graph Neural Networks". The authors could consider follow some of their experimental setup to strengthen the analysis part.*
>
> - We follow reviewer's suggestion to conduct deeper analysis as follows. To answer the question "whether the predicted operator per layer will be different", we split the operations in architectures into three groups according to their depths (the distance between the input and the layer), then we count the distributions of the occurrence of each operation at different depths. The results under setting $\beta=0.4$, $\delta=4.0$  are:
> | depth | GCN | GAT | GIN | ARMA | EDGE | MR | linear | skip |
> | -- | -- | -- | -- | -- | -- | -- | -- | -- |
> | 1 | 0.127 | 0.023 | 0.199 | 0 | 0.195 | 0 | 0.118 | 0.339 |
> | 2 | 0.379 | 0.092 | 0.130 | 0 | 0.215 | 0.023 | 0.057 | 0.103 |
> | $\ge 3$ | 0.280 | 0.085 | 0.203 | 0 | 0.390 | 0.025 | 0.008 | 0.008 |
>
> We can find that operations appear differently in different depth. E.g., at depth 1, skip-connect appears  most frequently, at depth 2, GCN appears most frequently, at depth $\ge 3$, edge_conv appears most frequently. All in all, non-message-passing operations (linear, skip) appear more for shallower positions, while message-passing operations appear more for in deeper position.
> - To answer the question "how this result is associated with some graph properties", we run the same experiment in different settings. We find that although the operation occurrence frequency is different, the rule "non-message-passing operations (linear, skip) appear more frequently for shallower position, while message-passing operations appear more frequently for deeper position" is basically held in all settings. We will include all these findings in our manuscript.
>
> *Comment 3: Such method cannot scale up to large-scale graph as the edge number goes quadratically to node size, and also it cannot be inductive to new nodes.*
>
> - In our algorithm, we only consider the edges existing in the original graph. The weights of non-existing edges are always 0. Therefore, the memory and time complexity are O(|E|).
> - Design an inductive method for new nodes in structure learning is a difficult task, which is not the focus of this paper.  We consider the ability of being inductive to new nodes as a future work. Thanks for the suggestion.
>
> *Comment 4: The authors only conduct experiments on three small node classification datasets. It would be more convincing to add results on some large-scale graph datasets, such as the ones in OGB, as the scale of network data is very important to the quality of both structure learning and NAS.*
>
> - We follow your suggestion and further conduct experiments on three larger graph benchmarks: Physics, CoraFull and OGB-arxiv. For Physics and CoraFull, we randomly split train/valid/test set as 50:25:25. For OGB-arxiv, we follow the default setting. The results show that our model still performs the best in larger graphs.  We will include all the experimental results  in our manuscript.
> | Dataset | Physics | CoraFull | OGB-arxiv |
> | -- | -- | -- | -- |
> | GCN | 95.94 | 68.08 | 70.39 |
> | GAT | 95.86 | 65.78 | 68.53 |
> | DARTS | 95.74 | 68.51 | 69.52 |
> | GASSO | 96.38 | 68.89 | 70.52 |

---

> > ### Comment · Reviewer_fXBu · 2021-08-23
> > **Comments after rebuttal**
> >
> > Thanks for authors' efforts to address my questions.
> >
> > The analysis and experiments are very convincing to me, so I would like to raise my rating.
> >
> > For the scalability issue, I agree that starting from given adjacency matrix is reasonable, but I suggest authors to discuss it and future directions, also some time cost statistics on large graphs (such as ogbn-arxiv).

---

> > > ### Author Response · Authors · 2021-08-24
> > > **Thanks for the reviewer's feedback.**
> > >
> > > We very much appreciate reviewer's feedback to our response.
> > >
> > > We will surely incorporate reviewer's suggestions into our revised versions.
> > >
> > >
> > > For the scalability issue, the reviewer's concern is reasonable, since considering densely connected edges needs $O(n^2)$ time/memory complexity. We think a possible future direction is to consider all edges in an incremental manner. On the basis of the given adjacency matrix, we gradually take the other most likely useful edges into consideration. We can control how many edges to be considered according to the time/memory capability. The more edges are considered, the fewer edges are missed, and the more accurate the structure is. We will add the discussion in our manuscript. As for the time cost statistics, we run our model ogbn-arxiv for 2.4s per epoch, we finish training for around 10min.

---

### Official Review · Reviewer_i3zt · 2021-07-16

**Rating:** 7
**Confidence:** 5

**Summary:**

This paper analyzes how DARTS selects its desired architectures, and shows that the gradient based NAS suffers from noises hidden in the graph, resulting in searching suboptimal GNN architectures. Besides, this paper improves the gradient based NAS methods by employing graph structure learning as a denoising process in the search procedure, and thus proposes GASSO, a more effective NAS algorithm. The theoretical analysis given by this paper is important and interesting, but the technical improvement lacks experimental analysis and proof, and important comparison experiments are missing.


**Limitations And Societal Impact:**

Yes

**Main Review:**

Strength
S1. This paper conducts theoretical analysis and measurement study with experiments to analyze how does the gradient based NAS method select its desired architectures.
S2. This paper analyzes and points out the shortcoming of the gradient based NAS methods, i.e., they suffer from noises hidden in the graph and thus result in searching suboptimal GNN architectures.
S3. The writing of this paper is good.

Weakness
W1. Important experiments are missing. This paper uses many paragraphs to explain and analyze the defect of DARTS, a gradient based NAS method, but does not compare with the proposed gradient based NAS method with it in Table 2. In the experimental part (Table 2), authors only compare GASSO with one RL-based NAS method, i.e., GraphNAS, and three GNN methods. Suggest to add more gradient-based NAS methods, e.g., GDAS, ASAP and XNAS, and state-of-the-arts GNN methods in the comparison experiments.
W2. Experimental datasets are not enough. Only three small datasets on GNN, i.e., Citeseer, Cora, PubMed, are used in the experiments. Suggest to add some large datasets, such as Physics and Corafull in the experiments.
W3. The motivation of the Hidden Feature Smoothness Constraint, and the experimental proof of its significance and rationality are missing and not clear. How does Hidden Feature Smoothness Constraint solve the defect of the gradient-based NAS methods, i.e., suffering from noises hidden in the graph and thus result in searching suboptimal GNN architectures? Suggest to add experiments to explain and analyze its function.
W4. What’s the definition of hidden feature mentioned in Section 4. Hidden Feature Smoothness Constraint is the key to solve the defect of the gradient-based NAS methods, suggest to give its definition.
W5. The details of the search space and the details of the searched GNN architectures are missing in the main paper.


**Time Spent Reviewing:**

10 hours

---

> ### Author Response · Authors · 2021-08-10
> **Response to Reviewer i3zt**
>
> Thank you for your reviewing efforts and constructive comments. We address reviewer’s concerns point by point.
>
> *Comment 1: Important experiments are missing. This paper uses many paragraphs to explain and analyze the defect of DARTS, a gradient based NAS method, but does not compare with the proposed gradient based NAS method with it in Table 2. In the experimental part (Table 2), authors only compare GASSO with one RL-based NAS method, i.e., GraphNAS, and three GNN methods. Suggest to add more gradient-based NAS methods, e.g., GDAS, ASAP and XNAS, and state-of-the-arts GNN methods in the comparison experiments.*
>
> - We have followed your suggestion and addressed your concern. We add DARTS, GDAS, ASAP, XNAS, DropEdge, FastGCN, GraphSAGE as our baselines. The results show that our model outperforms all used baselines. We will include all the experimental results including DARTS, GDAS, DropEdge, FastGCN, GraphSAGE,ASAP and XNAS  in our manuscript.
> | Dataset | Cora | Citeseer | Pubmed |
> | -- | -- | -- | -- |
> | DARTS | 86.18 | 79.10 | 86.47 |
> | GDAS | 85.41 | 74.20 | 89.50 |
> | ASAP | 85.21 | 75.14 | 88.65 |
> | XNAS | 86.80 | 76.33 | 88.61 |
> | FastGCN | 85.00 | 77.60 | 88.00 |
> | GraphSAGE | 82.20 | 71.40 | 87.10 |
> | DropEdge | 87.60 | 78.57 | 87.34 |
> | GASSO | 87.63 | 79.61 | 90.52 |
>
> *Comment 2: Experimental datasets are not enough. Only three small datasets on GNN, i.e., Citeseer, Cora, PubMed, are used in the experiments. Suggest to add some large datasets, such as Physics and Corafull in the experiments.*
>
> - We have followed your suggestion and addressed your concern. We conduct further experiments on three larger graph benchmarks: Physics, CoraFull and OGB-arxiv. For Physics and CoraFull, we randomly split train/valid/test set as 50:25:25. For OGB-arxiv, we follow the default setting. The results show that our model still performs well in larger graphs.  We will include all the experimental results in our manuscript.
> | Dataset | Physics | CoraFull | OGB-arxiv |
> | -- | -- | -- | -- |
> | GCN | 95.94 | 68.08 | 70.39 |
> | GAT | 95.86 | 65.78 | 68.53 |
> | DARTS | 95.74 | 68.51 | 69.52 |
> | GASSO | 96.38 | 68.89 | 70.52 |
>
> *Comment 3: The motivation of the Hidden Feature Smoothness Constraint, and the experimental proof of its significance and rationality are missing and not clear. How does Hidden Feature Smoothness Constraint solve the defect of the gradient-based NAS methods, i.e., suffering from noises hidden in the graph and thus result in searching suboptimal GNN architectures? Suggest to add experiments to explain and analyze its function.*
>
> - Through our differentiable graph structure mechanism, we can perform different aggregation operations for different nodes. E.g. a GCN layer can be represented as $\tilde{D}^\frac{1}{2}A\tilde{D}^\frac{1}{2}XW$, and an MLP layer can be represented as $IXW$. If we can change the structure part in the layer, i.e. $\tilde{D}^\frac{1}{2}A\tilde{D}^\frac{1}{2}$ in GCN, we can control different node to do different aggregation. In fact, $G$ in our model is a structure matrix between $\tilde{D}^\frac{1}{2}A\tilde{D}^\frac{1}{2}$ and $I$, which means some node aggregate more from neighbors while others are more like pass through an MLP layer. But how can we decide how nodes should aggregate? We use hidden feature smoothness. In condition that the model has learnt some knowledge, the model will give good predictions about node labels. Thus, the edge weights whose connected nodes have similar hidden features are larger than those have different hidden features. By updating the structure, similar node features are aggregated more in later GNN procedure, improving the denoising ability of the algorithm.
> - To better explain the mechanism, we study an experiment on the denoising ability in Section 5, beginning at line 353, and the experiment results are shown in Figure 3. The experiment shows that hidden feature smoothness constraint can decrease the weight of noisy edges, making GNN aggregate more useful information.
>
> *Comment 4: What’s the definition of hidden feature mentioned in Section 4. Hidden Feature Smoothness Constraint is the key to solve the defect of the gradient-based NAS methods, suggest to give its definition.*
>
> - Hidden feature is defined as the feature map after the softmax function in the model classifier, indicating the probability of which type the nodes belong to. The definition of hidden feature is described at line 269.
>
> *Comment 5: The details of the search space and the details of the searched GNN architectures are missing in the main paper.*
>
> - The candidate operations in our search space contain GCN, GAT, GIN, MRConv and linear, the number of operations varies according to the dataset (2 for Cora and CiteSeer, 4 for PubMed). The search space is described in Appendix B.2, beginning at line 52, we will move it to the main paper in the next version. As for the searched GNN architectures, since we directly use the super network in evaluation (mentioned at line 292), it may be difficult to show the searched architectures.

---

### Official Review · Reviewer_6Tss · 2021-07-16

**Rating:** 9
**Confidence:** 5

**Summary:**

This paper proposed one graph neural architecture search method to learn graph structures in one differentiable way. It makes an investigation on how NAS select the operations in GNNs, and then evaluate the NAS method on a set of synthetic datasets with different structures. This paper proposed one feature smoothness constraints which can learn the graph architecture in one differentiable way.

**Limitations And Societal Impact:**

yes

**Main Review:**

#### Pros:
1. This paper provides the theoretical analysis about how NAS method selects the operations in GNNs which is novel and interesting to understanding NAS.
2. The experiments on these synthetic datasets are interesting and it can verify that NAS can select data-specific architectures when facing different structures and features.
It is conceptual novel in understanding NAS in GNN architecture search.

#### Cons:
1. There is too little introduction about the constraints in section 4 thus it would be a little confusing to me. How to initialize G? The distance loss in Eq (2) aims to minimize the connections between inter-group nodes. It seems that the node labels are encoded in G_0 and we can identify whether one edge (i, j) belongs to the inter or intra group?

2. This paper learns to optimize the graph topology to obtain better performance. In exactly this, DropEdge [29] should be a natural baseline in experiments.

#### Other comments:
1. On those synthetic datasets, the relationship of operation selections and the different settings are shown. On the 3 real-world datasets, it would be better to show the \delta and \beta conditions, and how DARTS select operations on these 3 real-world datasets. I wonder whether the conclusion on real-world datasets is consistent with figure 2.
2. In line 130, I think it should be “the operation with the maximum \alpha of each edge is chosen”, not "the node".


**Time Spent Reviewing:**

5h

---

> ### Author Response · Authors · 2021-08-10
> **Response to Reviewer 6Tss**
>
> Thank you for your reviewing efforts and constructive comments. We address reviewer’s concerns point by point.
>
> *Comment 1: There is too little introduction about the constraints in section 4 thus it would be a little confusing to me. How to initialize G? The distance loss in Eq (2) aims to minimize the connections between inter-group nodes. It seems that the node labels are encoded in G_0 and we can identify whether one edge (i, j) belongs to the inter or intra group?*
>
> - We give each edge a parameter initialized as 0, and the weights are calculated from a sigmoid function of the parameter. Therefore, the initialized weights are 0.5 for all edges.
> - We guess the reviewer means G_o instead of G_0. Since all edge weights are 0.5 in G_o, we cannot identify whether one edge belongs to the inter or intra group.
>
> *Comments 2: This paper learns to optimize the graph topology to obtain better performance. In exactly this, DropEdge [29] should be a natural baseline in experiments.*
>
> - We follow your suggestion and run DropEdge+GCN as one of our baselines, the results are
> | Dataset | Cora | Citeseer | Pubmed |
> | -- | -- | -- | -- |
> | DropEdge | 87.60 | 78.57 | 87.34 |
> | GASSO | 87.63 | 79.61 | 90.52 |
>
> *Comment 3: On those synthetic datasets, the relationship of operation selections and the different settings are shown. On the 3 real-world datasets, it would be better to show the \delta and \beta conditions, and how DARTS select operations on these 3 real-world datasets. I wonder whether the conclusion on real-world datasets is consistent with figure 2.*
>
> - Thanks for the suggestion,  we will report the results in the updated version of our manuscript.
>
> *Comment 4: In line 130, I think it should be “the operation with the maximum \alpha of each edge is chosen”, not "the node".*
>
> - Thank you for the correction, we will modify the sentence in the next version.

---

### Official Review · Reviewer_Sygr · 2021-07-16

**Rating:** 6
**Confidence:** 4

**Summary:**


The paper investigates the ability of gradient-based NAS method on selecting desired operators to construct GNN architectures. It shows that DARTs method can discriminate the benefits of different operators based on the graph structure but suffers from the noised in graph. Based on the findings, the authors propose to add an extra graph-structure learning term during the optimization process to denoise the graph structure and shows improved performance compared to existing baselines.


**Limitations And Societal Impact:**

The author didn’t describe the limitations and potential negative societal impact.

**Main Review:**

Positive points:
1.	The paper is well-organized with easy-to-follow presentation.
2.	The idea of refining the architectures is novel. Analyzing the theoretical property and noise effect to NAS algorithms is of great value.

Comments:
1.	Theorem 1 seems to be intuitive and straightforward, and I don’t think the second theorem answers the question “how optimal are the architectures selected by graph NAS”. It only shows the linear operator are preferable than GCN under certain conditions, which I think it’s kind of overclaim. Also, it is hard to say how close the synthetic setting is to the real graph settings and there’re no experiments on real graphs to demonstrate this.
2.	The way of how W and A is updated based on the weighted edges is not clear. Please clarify. Also, it would be great to add more theoretical foundations rather than pure intuition of why the specific smoothness constraint is added.
3.	I’m curious whether adding the structure refining term would benefit other baselines. It would be great to add these experiments if possible.
4.	Bayesian optimization-based methods are not described in the second graph of the introduction. Also, some NAS for GNN references are not included such as Kaixiong Zhou, Qingquan Song, Xiao Huang, and Xia Hu. Auto-gnn: Neural architecture search of graph neural networks. arXiv preprint arXiv:1909.03184, 2019.
5.	I’m worried about the memory/time complexity of the proposed method. It is also great to do a complexity and convergence analysis for the final algorithm.
6.	It’s better to add a 2D plot to better described the joint effect of two variables in Figure 2.


**Time Spent Reviewing:**

3 hours

---

> ### Author Response · Authors · 2021-08-10
> **Response to Reviewer Sygr**
>
> Thank you for your reviewing efforts and constructive comments. We address reviewer’s concerns point by point.
>
> *Comment 1: Theorem 1 seems to be intuitive and straightforward, and I don’t think the second theorem answers the question “how optimal are the architectures selected by graph NAS”. It only shows the linear operator are preferable than GCN under certain conditions, which I think it’s kind of overclaim. Also, it is hard to say how close the synthetic setting is to the real graph settings and there’re no experiments on real graphs to demonstrate this.*
>
> - As for Theorem 1, we give a quantitative analysis of the metric of operation goodness, for the first time. It is easy to have a straightforward qualitative proposition like “good operations get larger weight”, but our analysis provides more accurate understanding of DARTS mechanism, such as that DARTS prefers operations who can help to correct the predictions on hard data.
> - Theorem 2 is not for answering the question “how optimal are the architectures selected by graph NAS”. We try to answer the question in the last part of Section 3 “Accuracy of Operation Selection”, beginning at line 223, and the corresponding experiment results are shown in Table 1. We propose Theorem 2 to show that different operations fit graphs with different amount of information in the node features and graph structures, preparing for following experiments.
> - The aim of using synthetic setting is to explore DARTS behaviors and answer the two questions by convenient simulation control of structure and node feature information in the graph, which can provide us with inspiration to design better Graph NAS algorithms.  We follow previous work [1] to conduct the synthetic experiment.
>
> [1] Wang, Xiao, et al. “Am-gcn: Adaptive multi-channel graph convolutional networks.” Proceedings of the 26th ACM SIGKDD International conference on knowledge discovery & data mining. 2020.
>
> *Comment 2: The way of how W and A is updated based on the weighted edges is not clear. Please clarify. Also, it would be great to add more theoretical foundations rather than pure intuition of why the specific smoothness constraint is added.*
>
> - As suggested by the reviewer, we will clarify this process in the main body part of the updated version. Actually we conduct gradient descent to W and A as shown in Appendix B.1. During the gradient descent, edge weights are considered in the calculation. E.g. GCNConv in PyTorch-geometric has implemented `edge weight` as a parameter in `forward` function, where node features are added with different weights in the aggregation phase. We apply the mechanism to all candidate operations in our implementation.
> - We will add more foundations as suggested by the reviewer as follows.
> Through our differentiable graph structure mechanism, we can perform different aggregation operations for different nodes. E.g. a GCN layer can be represented as $\tilde{D}^\frac{1}{2}A\tilde{D}^\frac{1}{2}XW$, and an MLP layer can be represented as $IXW$. If we can change the structure part in the layer, i.e. $\tilde{D}^\frac{1}{2}A\tilde{D}^\frac{1}{2}$ in GCN, we can control different node to do different aggregation. In fact, $G$ in our model is a structure matrix between $\tilde{D}^\frac{1}{2}A\tilde{D}^\frac{1}{2}$ and $I$, which means some node aggregate more from neighbors while others are more like pass through an MLP layer. But how can we decide how nodes should aggregate? We use hidden feature smoothness. In condition that the model has learnt some knowledge, the model will give good predictions about node labels. Thus, the edge weights whose connected nodes have similar hidden features are larger than those have different hidden features. By updating the structure, similar node features are aggregated more in later GNN procedure, improving the denoising ability of the algorithm.
>
> *Comment 3: I’m curious whether adding the structure refining term would benefit other baselines. It would be great to add these experiments if possible.*
>
> - We have done this study in the paper. We combine the structure refining mechanism with GCN and GAT (we call them GASSO-V6 and GASSO-V7 in the paper). Please refer to Ablation Study part in Section 5, beginning at line 324, and the experimental results are shown in Table 2.
>
> *Comment 4: Bayesian optimization-based methods are not described in the second graph of the introduction. Also, some NAS for GNN references are not included such as Kaixiong Zhou, Qingquan Song, Xiao Huang, and Xia Hu. Auto-gnn: Neural architecture search of graph neural networks. arXiv preprint arXiv:1909.03184, 2019.*
>
> - Thank you for the suggestion, we will add some introduction about Bayesian optimization-based methods and Auto-GNN in the next version.
>
> *Comment 5: I’m worried about the memory/time complexity of the proposed method. It is also great to do a complexity and convergence analysis for the final algorithm.*
>
> - We will add memory/time complexity analysis as follows. In our algorithm, we only consider those edges that exist in the original graph. The weights of non-existing edges are always set to 0. Therefore, the memory and time complexity are O(|E|). Empirically, our algorithm converges within 100~200 epochs.
>
> *Comment 6: It’s better to add a 2D plot to better described the joint effect of two variables in Figure 2.*
> - Thank you for the suggestion, we will add a plot to show our analysis of the joint effect of two variables in detail.

---

> ### Comment · Reviewer_Sygr · 2021-08-21
> **Review addition after the author's feedback**
>
> After reading the rebuttal, the author mostly addressed my concerns so I would like to raise my rating from 5 to 6.

---

### Decision · Program_Chairs · 2021-09-27

**Decision:**

Accept (Poster)

**Comment:**

This paper studied the NAS problem for GNNs, provided analysis on the limitation of existing gradient based NAS for this task, and proposed to perform graph structure learning at the same time. The reviewers generally find the analysis is interesting and the paper to be well written. The authors have provided additional experiments and explanations during the rebuttal, and all the reviewers found the rebuttal to be helpful during the committee discussion. Based on this, we recommend the acceptance of the paper.